# Phage Display Selection of an Anti-Idiotype-Antibody with Broad-Specificity to Deoxynivalenol Mycotoxins

**DOI:** 10.3390/toxins13010018

**Published:** 2020-12-28

**Authors:** Janne Leivo, Markus Vehniäinen, Urpo Lamminmäki

**Affiliations:** 1Department of Biochemistry, University of Turku, 20520 Turku, Finland; urplammi@utu.fi; 2Biovian Ltd., 20520 Turku, Finland; markus.vehniainen@biovian.com

**Keywords:** antibody library, phage display, mycotoxin, deoxynivalenol, immunoassay

## Abstract

The use of synthetic antibody libraries and phage displays provides an efficient and robust method for the generation of antibodies against a wide range of targets with highly specific binding properties. As the in vitro selection conditions can be easily controlled, these methods enable the rapid generation of binders against difficult targets such as toxins and haptens. In this study, we used deoxynivalenol mycotoxin as a target to generate anti-idiotype-antibodies with unique binding properties from synthetic antibody libraries. The binding of the selected anti-idiotype antibodies can be efficiently inhibited with the addition of free isoforms of deoxynivalenol. The antibody was consecutively used to develop deoxynivalenol-specific ELISA and TRF-immunoassays, which can detect deoxynivalenol and two of the most common metabolic isoforms in the range of 78–115 ng/mL.

## 1. Introduction

Synthetic antibody libraries have been proven to be a powerful tool for the development of binders against low-molecular weight compounds [1]. Although immunization-based methods are still considered to be the “golden standard” of antibody development, the use of synthetic antibody libraries as a source of binders circumvents some of the most common problems in the antibody generation process, mainly related to the toxicity or lack of immunoresponse [2]. In addition, as the selection conditions of the antibody generation process can be easily controlled, it provides a rapid method for the development of antibodies with various binding properties. The intrinsic antigenicity of antibodies makes it possible to generate binders which recognize regions of another antibody [3]. These anti-idiotype antibodies (anti-Id-Ab) are particularly useful when the anti-idiotypic interaction is located in the proximity of the antigen-binding site. Anti-Id-Abs, which either interfere or are dependent on the presence of the free antigen, have been applied for the detection of various molecular interactions without the need of chemically modified analytes [4,5,6]. 

As the interaction is primarily based on the binding of the two antibodies, anti-id-Abs have been found to be especially useful in the detection of toxic compounds, such as mycotoxins [7] and cyanotoxins [8]. In this study, deoxynivalenol (DON) was used as a model antigen for the generation of anti-Id-Abs with phage display selections. Deoxynivalenol belongs to a group of structurally diverse mycotoxins mainly produced by Fusarium species molds (Figure 1). These mycotoxins, more specifically termed trichothecenes, at high concentrations create a food safety hazard to humans and animals as they are a common contaminate in field crops and dry food stuffs [9]. In addition to the health hazards, mycotoxins decrease the overall yield and quality of crops, and as a result create financial losses for the food producers [10]. The removal of trichothecenes from infected food products is difficult due to the small size and high stability of DON and its metabolic derivatives 3-acetyldeoxynivalenol (3-AcDON) and 15-acetyldeoxynivalenol (15-AcDON). This creates the need for a reliable and efficient detection method which is suitable for field testing [11]. The development of such a simple and affordable assay for the control of trichothecene levels in foodstuffs would create economic benefits for the food manufacturers, and subsequently increase the overall quality of the affected food products.

We describe the generation of anti-Id-Ab, which has specificity also towards the free antigen. This broad-specificity antibody was used to develop an immunoassay using two different detection methods and is capable of detecting DON and its most common metabolic isoforms in the range of 78–115 ng/mL.

## 2. Results and Discussion

This study describes an efficient method for the generation of anti-Id-Abs with novel binding properties from the synthetic antibody repertoire. Surprisingly, the binding of the found anti-Id-Abs can be inhibited with the use different isoforms of the antigen. An existing monoclonal antibody (10B5) in a complex with the free antigen deoxynivalenol was used as the target in phage display selections. The selection pressure was directed towards the recognition of the antigen binding site of the 10B5 antibody with the use of depletive steps prior the addition of free DON to the selections. The aim was to generate specificity solely towards the regions involved in the antigen-binding interaction. After three selection rounds, antigen-specific enrichment could be observed from the phage stock in a TRF-immunoassay (Figure 2a). From the third-round antibody population, 95 individual clones were screened, of which 72 (76%) were target (antibody/DON)-specific. In addition, with 19 (26%) of the target-specific clones, the binding interaction of the primary (10B5) and the screened anti-Id-Ab was inhibited >75% in the presence of soluble DON (500 ng/mL). We selected one antibody (cDON_1) with the most interesting binding properties for further analysis (Figure 2b). Interestingly, the cDON_1 can recognize DON and the two metabolic isoforms, 3-AcDON and 15-AcDON, within the same affinity range of 78–115 ng/mL (Figure 2c,d, Table 1). However, there is no specificity for the structurally related trichothecenes compounds, such as nivalenol (NIV), T-2 or HT-2 toxin. Based on the immunoassay data and the structures of the common type A and B trichothecenes, the cDON_1 antibody interacts with the antigen around the position C-4 (R2 in Figure 1). The binding of DON or its isoforms to the antibody induces conformational changes in the structure of cDON_1, which prevent the interaction with 10B5. The DON-cDON_1 interaction can be confirmed by the cross-reactivity profiles of the two antibodies: 10B5 does not detect the 15-AcDON isoform, which in turn inhibits the binding of cDON_1 with equal efficiency compared to the structurally related compounds DON and 3-AcDON (Figure 2c,d).

In related studies, the inhibition of the anti-idiotypic binding interaction has been described to be based either on the blocking of the binding site [12], or on molecular mimicry of the target antigen [13]. However, the inhibition of the 15-AcDON isoform confirms the 10B5-cDON_1 interaction to be more complex, and both antibodies to be antigen-specific. This creates an interesting possibility for the generation of antigen-specific binders without the laborious chemical conjugation and purification of the small target antigens. In addition, with the lack of chemical conjugation reactions, which are often carried out in harsh conditions, this method preserves the native structure of the antigen and is also suitable for chemically labile molecules. Furthermore, this method could enable the development of antibodies also against epitopes that are unavailable due to, i.e., a lack of conjugation sites. The carrier protein specificity, a well-recognized problem for low-molecular weight compounds [14], does not seem to interfere with the binding interaction of DON and cDON_1 in the assay described here.

Anti-Id-Abs have been proven to be highly useful in the development of immunochemical application for the detection of low-molecular weight contaminants from environmental and food samples. Anti-Id-Abs have also been applied to develop extremely sensitive immunoassays for the detection of haptenic structures [15]. The toxicity and low-molecular weight of trichothecenes makes them a challenging target for the development of antibodies with immunization-based methods. Anti-Id-Abs are an interesting alternative for the detection of such toxic, low-molecular weight targets. However, the generation of anti-Id-Abs remains challenging, and often requires the direct conjugation of the antigen to the target antibody.

We describe a method for the rapid development of anti-Id-Abs, as well as a novel detection method of the deoxynivalenol-specific antibody, deoxynivalenol, and its most common derivatives based on an anti-idiotype antibody derived from the synthetic binder repertoire. The binding interaction of the two antibodies can be subsequently inhibited with the addition of free isoforms of DON.

## 3. Materials and Methods

### 3.1. Materials and Reagents

DELFIA series buffers, streptavidin and rabbit anti-mouse-coated microtiter plates were purchased from Kaivogen Diagnostics (Turku, Finland). All measurements were done with a Victor 1420–fluorometer from Perkin-Elmer (Turku, Finland). The magnetic nanoparticles and magnetic bead concentrator were purchased from Dynal (Norway). The mycotoxins nivalenol (NIV), Deoxynivalenol (DON), 3-Acetyldeoxynivalenol (3-AcDON), 15-Acetyldeoxynivalenol (15-AcDON), T-2 toxin and HT-2 toxin were purchased from Biopure Guntramsdorf, Austria). Hyperphages were obtained from Thermo Fisher Scientific (Waltham, MA, USA). The *E. coli* cell lines used for the sorting and expression of the antibody libraries were purchased from Stratagene (La Jolla, CA, USA): BL21 (F-, dcm, ompT, hsdS[rB− mB−], gal [malB+], K-12[λS]) and XL1-Blue (recA1, endA1, gyrA96, thi-1, hsdR17, relA1, lac [F`, TetR]). All microbiological reagents were prepared as described in Sambrook et al. [16]. The single-chain alkaline phosphatase (scFv-ALP) fusion proteins were purified with HisPur Ni-NTA spin columns Thermo Fisher Scientific (Waltham, MA, USA). The ELISA substrate para-nitrophenylphosphate (pNPP) was obtained from Sigma-Aldrich (St. Louis, MO, USA).

### 3.2. Biotinylation of Antibodies

The capture antibodies used in the immunoassays, 10B5 (100 µg) and cDON_1 (500 µg), were mixed with the EtOH (99.5%) solution containing a 50× molar excess of biotin isothicyanate (BITC, University of Turku). The pH of the reaction was adjusted with carbonate buffer (0.5 M, pH 9.8) and incubated for four hours at RT. The excess biotin was removed with two consecutive purifications through the NAP-5 column (Amersham Bioscience, Buckinghamshire, UK). The concentration of the protein was determined with Bradford reagent (BioRad, Hercules, CA, USA).

### 3.3. Phage Display Selections

The synthetic antibody libraries used for the phage display selections have been described in previous studies by Brockmann et al. [17] and Huovinen et al. [18]. The mouse monoclonal antibody, 10B5, specific to DON, was a kind gift from Professor Christopher Elliot, Queens University, Belfast, United Kingdom. The binding properties of the 10B5 in Fab format have previously been described in Romanazzo et al. [19]. The phage display selections were carried out with the following conditions: The biotinylated 10B5 IgG (20 µg) was bound to M280 Streptavidin beads supplemented with DON (0.1 µg) for 1 h in rotation. The beads were washed three times with TBT-0.1 buffer and 1 × 10^12^ tfu of library phages were mixed with beads. Soluble mouse IgG (100 µg) unspecific to DON was added to the reaction to deplete all phages binding to regions irrelevant to the antigen binding. The reaction was incubated for 3 h at RT. The beads were washed five times with TBT-0.1 buffer and once with TSAT before elution with 10 µg/mL of trypsin for 30 min at RT. The eluate was used to infect XL1-Blue cells in the exponential growth phase. The phages were repropagated from the cells collected from the output plate as described in Ref. [20]. A total of three selection rounds were performed whereby the amount of antigen was reduced to half after each round.

### 3.4. Antibody Screening and Characterization

The screening of individual antigen-specific antibodies was done in single-chain fragment variable (scFv) format, where the scFv was displayed on the surface of the M13 bacteriophages. Individual colonies (*n* = 95) were picked from the output plate of the third selection round. The cells were grown on a 96-well tissue microtiter plate in a 150 µL volume of Super Broth (SB) supplemented with 25 µg/mL of chloramphenicol, 10 µg/mL tetracycline and 0.1% glucose at +37 ˚C, 900 rpm for 6 h. After the incubation, 1 × 10^10^ tfu/mL of hyperphages were used to infect the cells for 30 min at 37 °C without shaking. The overnight production of the phage-antibodies was done in a plate shaker set at 900 rpm at +26 °C. The cells were removed with 4000 rpm centrifugation for 15 min at +4 °C and 5 µL of the culture supernatant was used in the primary screening immunoassay. The phagemid vector (pEB32x) was isolated from the antibodies showing the desired binding properties with a Qiagen DNA miniprep kit according to the manufacturer’s protocol. The isolated scFv-genes were cloned to the pLK06H and pLK04 [18] vectors for the soluble expression of scFv-BALP (single-chain fragment variable fused to bacterial alkaline phosphatase) and scFv proteins. The antibodies were produced in a 100 mL culture of Bl21 cells as described previously [21]. The scFv fragment of the cDON_1 clone was produced in a larger scale fermentation in 4 L of SB and purified as described previously [22].

### 3.5. Immunoassays

The immunoassay steps were carried out in 100 µL volume with slow shaking for 1 h at room temperature and with four washes between each step. All samples were taken in duplicates and the time-resolved fluorescence europium signal was measured with a Victor 1420–fluorometer.

Immunoreactivity: The follow-ups on the enrichment of the antigen-specific phage populations were performed on streptavidin plates with 20 ng/well of biotinylated 10B5-Mab. After washes, the scFv-displaying phages were added in three replicas of 1 × 10^8^ tfu/well with (500 ng) and without free DON. The bound phages were detected with europium-labeled anti-VCSM13 antibody (5 ng/well).

Screening: The individual antibodies (scFv) displayed on phages were analyzed with a competitive immunoassay. First, streptavidin-coated microtiter wells were coated with 20 ng of biotinylated 10B5-Mab. After washes, 10 µL of overnight-produced phage culture was mixed with 90 µL of assay buffer containing either 0 ng or 500 ng of DON. From the signal level difference, the inhibition (%) was determined for all individual clones. Based on the results we selected one clone (cDON_1) which had an inhibition of 96% in the presence of DON for further studies.

ELISA: The competitive immunoassay for the characterization of the cDON_1 antibody was performed as follows: biotinylated 10B5 (20 ng/well) was bound to streptavidin-coated microtiter strips. After the washes five different concentrations ranging from 10 to 5000 ng/mL of soluble antigen (DON, 3-AcDON, 15-AcDON, NIV, T-2 and HT-2) were added to the wells in combination with the reporter antibody. After the final washes, the colorimetric reaction of BALP activity was carried out with the use of pNPP and the absorbance at 405 was measured with a Victor 1420–fluorometer.

TRF-IA: The characterization of the cDON_1 antibody with TRF-IA was done as follows: biotinylated cDON_1 (10 ng/well) was bound to streptavidin-coated microtiter strips. After the washes five different concentration points ranging from 1 to 10000 ng/mL of soluble antigen (DON, 3-AcDON, 15-AcDON, NIV, T-2 and HT-2) were added to the wells in combination with the DON-specific monoclonal antibody 10B5. The bound 10B5 was detected with the use of europium-labeled rabbit anti-mouse (RAM, 10 ng/well) IgG. The TRF europium signal was measured with a Victor 1420–fluorometer.

### 3.6. Data Management

The enrichment of the antigen specific phage-antibodies between each selection round was inferred from the signal to background levels of the TRF-assay. The IC_50_ values for the assays were determined with the use of Graphpad Prism 7 (GraphPad Software Inc., San Diego, CA, USA).

## Figures and Tables

**Figure 1 toxins-13-00018-f001:**
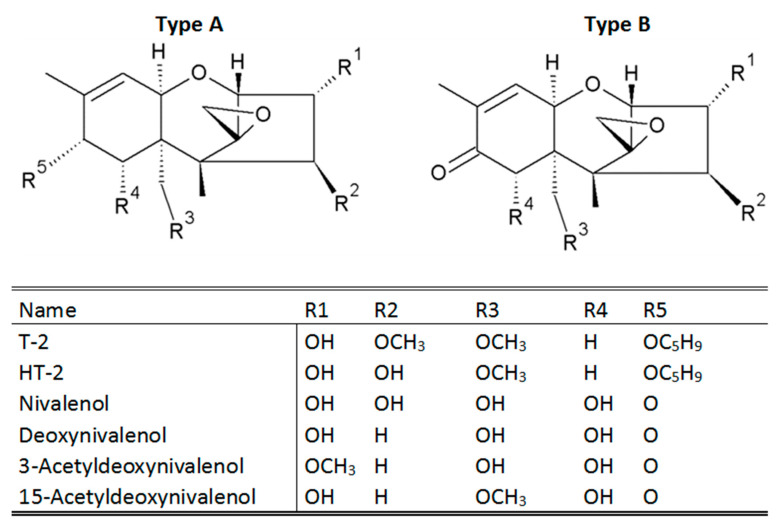
The structures of type A and B trichothecenes used in this study.

**Figure 2 toxins-13-00018-f002:**
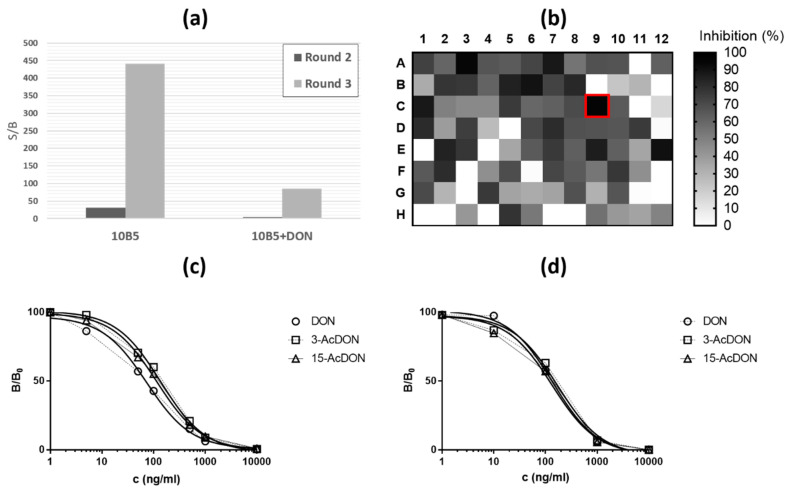
(**a**) Immunoreactivity of the phage pools isolated after selection rounds 2 and 3. The y-axis depicts the ratio of the TRF europium signal to streptavidin background (S/B) with and without the presence of free DON (500 ng). (**b**) Screening of individual anti-Id-Abs with a competitive immunoassay on a 96-well plate. The inhibition (%) with free DON was used to select the antibody cDON_1 (C09, highlighted). (**c**) Inhibition curves of competitive ELISA using 10B5 Mab as a capture and the scFv-BALP (cDON_1) as a tracer. Variable concentrations of DON, 3-AcDON and 15-AcDON. (**d**) Inhibition curve of a competitive TRF-IA, where cDON_1 was used as a capture and 10B5 Mab as a tracer with variable concentrations of DON, 3-AcDON and 15-AcDON.

**Table 1 toxins-13-00018-t001:** Cross-reactivity profile of the 10B5 and cDON_1 antibodies used in sandwich immunoassays (ELISA and TRF-IA).

	cDON_1				10B5 *	
	ELISA		TRF-IA			
Target	Reactivity	IC50 (ng/mL)	Reactivity	IC50 (ng/mL)	Reactivity	IC50 (ng/mL)
DON	X	77.7	x	128	x	107
3-AcDON	X	111.5	x	168	x	123.8
15-AcDON	X	114.9	x	135	-	-
NIV	-	-	-	-	-	-
T-2	-	-	-	-	-	-
HT-2	-	-	-	-	-	-

* Values obtained by electrochemical immunosensor with monoclonal DON-specific antibodies.

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
