# Peer review of "Phage Display Selection of an Anti-Idiotype-Antibody with Broad-Specificity to Deoxynivalenol Mycotoxins"

_toxins, 2020, doi:10.3390/toxins13010018_

Round 1

Reviewer 1 Report

The authors have reported the set up of highly sensitive TRF and ELISA based assays for detection of deoxynivalenol (DON). The assays have been performed using the monoclonal antibody 10B5 and a new specific anti-idiotype antibody derived from scFv phage display screening.

The authors have mentioned the use of monoclonal antibody 10B5 IgG referring to Romanezzo et al ( reference number 19) however, in this article Fab fragment rather than full-lenght antibody has been used. The correct reference is: T. Korpimäki, V. Hagren, E.-C. Brockmann, M. Tuomola Anal. Chem., 76 (11) (2004), pp. 3091-3098, that corresponds to reference number 22. The authors should specify if they have used mAb or recombinant / proteolitic Fab fragment.

Generally, the authors  are not clear about the nomeclature related to the antibody formats that they used. For example, the Paragraph 3.4 is related to scFv screening not to full-lenght antibody. In the same paragraph, the reference number 22 is not right. The authors should insert some suitable references in which the expression and purification of scFv are reported. A more detailed description of these procedures and related results are required as supplementary informations, at least.

The screening is not well-presented. The selection of scFv clone, named c-DON1, is not argued. In the methods section is required the association of data related to screening of 95 scFv clones. For instance, the introduction of a table including the best 10 clones and related TRF assays data is suggested. A more detailed description in the caption of Figure 2a is required. For example, the specification of use of europium labelled anti-VCSM13 as detection system.

In the Figure 2b, the authors have reported the data obtained from competitive ELISA assay whereas the results about TRF-IA are missing. This is also not clear. The competition assays have been carried out on 95 scFV clones or only on the selected cDON1 clone? In the caption of Figure 2b, the authors have indicated the use of c_DON1 antibody but in the materials they have reported the measurements of BALP activity using pNPP substrate. This means that the authors in ELISA assay have used scFv not cDON1 antibody. On the other hands, the authors described the use cDON1 antibody in the TRF-IA assay. The authors should include the data related to this TRF-IA assay. Finally, a more detailed explanation about analysis data is required especially related to TRF measurements supporting by references.

Reviewer 2 Report

Excellent piece of work where authors identified anti-idiotype antibodies that are the basis for development of highly sensitive assay for detection of a mycotoxin.

Two typographical errors: 

ELISA sustrate

libraries used for the phage display selections has been described

Unclear sentence:

 The step was supplemented with ...

Was the step supplemented, or something else?

Author Response

We thank the reviewer for the comments. The typographical errors and unclear sentences have now been corrected in the manuscript.

Round 2

Reviewer 1 Report

The paper is now accepted for publication.